# Identification of Individuals of Two Takin Subspecies Using Biological and Ecological Criteria in Eastern Himalayas of China

**DOI:** 10.3390/ani14162426

**Published:** 2024-08-21

**Authors:** Yuan Wang, Yonglei Lv, Guanglong Wang, Feng Liu, Yingxun Ji, Zheng Liu, Wanglin Zhao, Wulin Liu, Pu Bu Dun Zhu, Kun Jin

**Affiliations:** 1Ecology and Nature Conservation Institute, Chinese Academy of Forestry, Beijing 100091, China; wangyuan@caf.ac.cn; 2Tibet Autonomous Region Research Institute of Forestry Inventory and Planning, Lhasa 850000, China; lvyonglei@ioz.ac.cn (Y.L.); 15065646456@163.com (G.W.); liufeng01@163.com (F.L.); wulinliu01@163.com (W.L.); pubudunzhu0430@163.com (P.B.D.Z.); 3Research Institute of Natural Protected Area, Chinese Academy of Forestry, Beijing 100091, China; 4Key Laboratory of Biodiversity Conservation of National Forestry and Grassland Administration, Beijing 100091, China; 5University of Chinese Academy of Sciences, Beijing 100000, China; 6Key Laboratory of Zoological Systematics and Evolution, Institute of Zoology, Chinese Academy of Sciences, Beijing 100083, China; 7Forestry and Grassland Bureau of Linzhi City, Tibet Autonomous Region, Linzhi 860010, China; grus0717@126.com; 8Forestry and Grassland Bureau of Medog County, Tibet Autonomous Region, Medog 860799, China; liuzheng1988@163.com; 9Medog Earth Landscape and Earth System Comprehensive Observation and Research Center, Chinese Academy of Sciences, Medog 860799, China; zhaowanglin@itpcas.ac.cn

**Keywords:** takin, camera trapping, MaxEnt niche model, Eastern Himalayas, biodiversity hotspots

## Abstract

**Simple Summary:**

Takins in the Eastern Himalayas of China are very diverse; however, species research in this region is lacking. Therefore, limited background data are available on the different subspecies present in this region. Using ecological models and surveys, this study examined the factors of two subspecies in this region, including phenotypic characteristics, population, activity, and distribution range, to provide species identification and classification references. The results revealed that the Mishmi takin and Bhutan takin could be differentiated using camera trapping. Their distribution areas, population densities, and population sizes could be determined in all human-accessible areas of the Eastern Himalayas in China. Thus, this study contributes to basic animal diversity knowledge and provides detailed information and references for species identification, distribution ranges, and population characteristics of the Mishmi and Bhutan takins. Moreover, this study provides resource data for subspecies classification that can be used to promote effective protection measures for endangered species.

**Abstract:**

Limited background data are available on the Mishmi takin (*Budorcas taxicolor taxicolor*) and Bhutan takin (*Budorcas taxicolor whitei*) subspecies in the Eastern Himalayas of China because of the lack of systematic field investigations and research. Therefore, mature-animal ecological methods were used to evaluate these takin subspecies’ phenotypic characteristics, distribution range, activity rhythm, and population size. From 2013 to 2022, 214 camera traps were installed for wild ungulate monitoring and investigation in all human-accessible areas of the Eastern Himalayas, resulting in 4837 distinguishable takin photographs. The external morphological characteristics were described and compared using visual data. Artificial image correction and related technologies were used to establish physical image models based on the differences between subspecies. MaxEnt niche and random encounter models obtained distribution ranges and population densities. Mishmi takins have a distribution area of 17,314 km^2^, population density of 0.1729 ± 0.0134 takins/km^2^, and population size of 2995 ± 232. Bhutan takins have a distribution area of 25,006 km^2^, population density of 0.1359 ± 0.0264 takins/km^2^, and population size of 3398 ± 660. Long-term monitoring data confirmed that the vertical migration within the mountain ecosystems is influenced by climate. Mishmi takins are active at 500–4500 m, whereas Bhutan takins are active at 1500–4500 m. The two subspecies were active at >3500 m from May to October yearly (rainy season). In addition, surveying combined with model simulation shows that the Yarlung Zangbo River is not an obstacle to migration. This study provides basic data that contribute to animal diversity knowledge in biodiversity hotspots of the Eastern Himalayas and detailed information and references for species identification, distribution range, and population characteristics.

## 1. Introduction

Due to global differences in species composition and variations in local animal and plant resource survey intensity, direct biases are observed in species recognition. However, limitations in species research in certain regions limit our understanding of the animal world [1,2]. Scientists are increasingly using various new methods of identification (acoustic, photographic observation) and monitoring of animals, depending on their lifestyle characteristics [1,2,3,4]. A species is the most basic taxonomic unit in natural sciences. For rare and endangered species, incorrect species definitions lead to unreasonable protection measures, resulting in less effective protection of endangered species [5]. Furthermore, attempts to clarify the controversy associated with species and subspecies classification aim to reveal the evolutionary history, genetic potential, and potential molecular mechanisms underlying the adaptive evolution of a species and provide scientific guidance for the protection and management of species and subspecies, which are currently insufficient and must be promoted. Thus, ambiguity in species knowledge is caused by limited and prolonged but incomplete investigations [1,2].

The genus *Budorcas* belongs to the family Bovidae of the order Artiodactyla and includes typical large mountain ungulates. Taxonomists have traditionally proposed controversial classification schemes for intra-genus species based on evidence from various aspects, such as phenotypic characteristics, molecules, and regions [6,7]. This genus is primarily distributed in Shaanxi, Gansu, Sichuan, Xizang, Yunnan, and other regions of China, as well as in India, Nepal, Bhutan, and Myanmar. It is found largely in the Eastern Himalayas and eastward of mainland China, including eastern China, the Himalayas, Gaoligong Mountain, Liangshan Mountain, Qionglai Mountain, Minshan Mountain, and the Qinling Mountains [7,8,9,10]. Four subspecies of *Budoras* are distributed in China, and the morphological characteristics of the Mishmi and Bhutan takin subspecies distributed across the Eastern Himalayas of Xizang, China, are not obvious. The colors of the juveniles, sub-adults, and adults are prone to changes. In addition, the two subspecies are sympatric in the same region; therefore, they are relatively difficult to identify and study in the field [5,6,7,8,9,10]. However, long-term field investigations have shown subtle differences in these subspecies’ morphology and adaptive habitat strategies. A detailed study of species, subspecies taxonomic units, and interspecific morphology can provide a better understanding of the species in nature [1,2].

Yang et al.’s work [5] and other molecular-based studies found that incorrect species definitions lead to unreasonable conservation measures, affecting the effectiveness of endangered species protection. Various sequencing strategies have confirmed that takins are members of the Bovidae and Sheep subfamilies, supporting its taxonomic status as a sheep subfamily. Based on genetic evidence, the Mishmi takin (*B. taxicolor*) and Bhutan takin (*B. whitei*) subspecies were merged to form the Himalayan takin (*B. taxicolor*). In contrast, the Qinling takin (*B. bedfordi*) and Sichuan takin (*B. tibetana*) subspecies were merged into two independent species of Chinese takin (*B. tibetana*), with the geographical boundary between the two being the confluence area of the Nu, Lancang, and Jinsha Rivers. The above reports contradict the previous classification views of one, three, or four species; this unifies classification bias at the molecular level and promotes research and development based on molecular biology [5].

However, many doubts remain regarding the macroscopic characteristics, including appearance, quantity, and distribution range or the definition of mixed areas occupied by takins in the Eastern Himalayan Mountains.

## 2. Methods

### 2.1. Research Area

The Eastern Himalayan district is located southeast of Xizang. Its administrative scope covers six counties and districts in Linzhi City, including Bayi District, and Milin, Bomi, Langxian, Chayu, and Medog Counties, and three counties in Shannan City, including Longzi, Cuona, and Cuomei Counties, covering two cities, eight counties, and one district. The geographical coordinates are 28°0′58”–28°33′31” N and 91°29′47”–97°38′59” E, with a total area of 1.3 × 105 km^2^. The Eastern Himalayan district begins at the Danbaqu River Basin in Medog County to the east. It ends at the Niangjiangqu Valley in Cuona County and the Lacang area in Nuqu and Loza Counties to the west. The main mountain range is relatively low compared with the Himalayas, with an average elevation of <6000 m. The peaks of Nanjiabawa (7782 m) and Galabailei (7294 m) were observed in this area.

The annual precipitation east of the Yarlung Zangbo River Valley exceeds 2000 mm. The precipitation to the west of the Baoluoli River (Kamen River) in Cuona County, Shannan City, is lower, ranging from 1000 to 1500 mm. Therefore, the distribution of vegetation is considerably different from that of the natural area in the Chayuqu River Basin, with differences between the eastern and western sections of the area.

The tropical evergreen rainforest in the eastern section is distributed along the low mountain foothills and plains. In the lower reaches of the Yarlung Zangbo River, Xirang Village in Medog County (500 m above sea level) can be reached. Below an altitude of 1100 m is a low-mountainous tropical rainforest zone found on banks or some slopes of canyons. Tropical plants, such as *Dipterocarpus* sp., *Tetrameles nudiflora*, *Terminalia myriocarpa*, and *Lagerstroemia minuticarpa*, are distributed there.

The mountainous subtropical evergreen broad-leaved forest belt is 1100–2300 m above sea level and comprises *Castanopsis fargesii*, *Quercus glauca*, *Alsophila* sp., *Raphidophora decursiva*, *Manglietia caveana*, *Talauma hodgsonii*, and *Mussaenda decipiens*. It is humid and rainy, with the characteristics of a “moss forest”.

The mountainous warm temperate mixed coniferous and broad-leaved forest zone has an altitude of 2300–2900 m. Its main distributions include *Tsuga dumosa*, *Abies* sp., and *Larix* sp. forests.

The mountainous cold-temperate dark coniferous forest zone has an altitude of 2900–3800 m. It is mainly covered in subalpine shrubs, such as *Tsuga dumosa* forest, *Abies* sp. forest, *Acer* sp., and *Rhododendron* sp. These two types of forest belts are mostly distributed in the wide U-shaped valley area upstream of the branch ditch on glacial moraine hills.

The altitude of the alpine cold zone shrub meadow zone is 3800–4200 m. The meadow mainly comprises *Juncus* sp., *Poa* sp., *Carex* sp., *Polygonum* sp., *Primula* sp., *Gentiana namlaensis*, *Meconopsis Vig*., and *Pedicularis* sp. The shrubs comprise *Rhododendron recumbens* (*R. rep*), *Salix* sp., and *Cassiope* sp. Because of the relationship between altitude and temperature, few vegetation groups are distributed in the alpine cold weathering zone at 4200–4800 m and the alpine ice and snow zone above 4800 m.

Summarily, the terrain and landforms in the region are unique and encompass all natural zoning types visible on Earth. Plant sources are abundant, and their composition is complex and variable. Animal and plant survival complement each other with extremely high diversity, thus providing abundant food sources and suitable space for wild animal habitats; thus, the study region has one of the most abundant ungulate populations worldwide. According to incomplete statistics, this region has 30 species and subspecies of ungulates (Figure 1) [1,2,11,12,13].

### 2.2. Camera Trapping

Twelve representative areas with different altitudes and vegetation types within the study area were selected as survey areas, and camera-trapping survey technology was used to detect and record takin species in each area. The survey periods were from January 2013 to July 2013; October 2013 to May 2014; October 2016 to April 2017; October 2018 to July 2020; July 2020 to December 30, 2021; and March 2021 to March 2022. A total of 214 infrared cameras were used, covering an altitude of 582–4700 m, with a cumulative survey workload of 58,505 camera working days (Table 1).

The camera model used was a Dongfa”g Ho’gying E1C (Shenzhen Dongfang Hongying Technology Co., Ltd., Shenzhen, China), which was set to continuously capture three photos and one 15 s video after each trigger with the sensitivity set at “medium” and photo imprinting turned on. Each camera was equipped with a 32 GB SD memory card and 12 No. 5 rechargeable batteries with a capacity of 2800 mAh. No physical concealment was performed during camera installation, and no food bait or attractants were distributed. The field data record card was filled and numbered after placement [1,2].

The obtained photos were sorted, and folders comprising takin photos were established according to the different monitoring areas and elevations. Classification times were marked for a statistical analysis. The takins’ body shape, size profile, coat color, and physical behavior were recorded consecutively with camera trapping from the same position and at similar times or positions to determine whether they belonged to the same sessions. Photographs captured at different times were defined as independent photographs [1,2].

### 2.3. Camera-Trapping Data Processing

#### 2.3.1. Principle of Camera-Trapping Data Identification 

Based on long-term field investigations and experience in deploying infrared cameras, we determined the prerequisites for camera-trapping photo identification; that is, the same camera should be used for imaging during the day, and visually good photos should be used for identifying species, rather than photos or videos with poor nighttime imaging, to reduce human error caused by identification errors. Comparing the images captured by two cameras, A and B, positioned along the same line and separated by a distance of over 500 m, can be used to determine whether the same takin has passed by.

#### 2.3.2. Camera-Trapping Independent Site Full-Event Observation Method 

For the camera-trapping set in the field, when collecting data for processing, we summarized it into two types of events. In the first type of event, when all photos and videos were observed, it was evident that the camera was set within the home ranges of wild animals because the captured animals were mostly engaging in behaviors such as feeding, playing, decorating, mud bathing, water drinking, and resting. Wild animals captured using this type of camera trapping are relatively single and fixed. For evening imaging photos, when identifying wild animals, some photos can be inferred based on daytime activity habits, behavior, calls, and other animal aspects in the area. The second type of event can be observed in all photos and videos. The camera was set at the “crossroads” of wildlife activities and captured various animals and many activities. The captured animals were mostly imaged performing behaviors such as passing, chasing, playing, and observing. Based on identifying wild animals, researchers should have relatively annual, complete, professional, and long-term field observational foundations.

To minimize human error during later data identification and processing, we used five or more infrared cameras and three or more handheld single-lens reflex cameras in our long-term investigation and accumulated visual photo data over a long period to eliminate color differences in photo data caused by different instruments, environmental conditions, climatic factors, and other factors.

#### 2.3.3. Camera-Trapping Wildlife Identification Method 

Detailed observations of the colors, spots, and patterns of the head, ears, back, limbs, tail, and other body areas were performed. Photographs of different areas were compared and confirmed repeatedly. Based on the practical experience gained from activities such as wildlife rescue and judicial appraisal, we gradually increased our understanding of wildlife. We continuously compared the actual investigations of wild animal entities and infrared camera data to improve the level of observation and understanding of wild animals. Other materials, such as fur, carcasses, and feces, were used as supporting evidence.

### 2.4. AI Image Correction Technology

Here, we applied self-trained machine learning because, at the pixel level of an image, similar animals in different postures or environments may produce very different images; however, different groups of animals in similar positions or backgrounds will produce very similar images. We used physical and photo data collected each time as self-reinforcing fragments combined into a dataset to overcome this error. Continuous positive feedback formed the final framework of the animal model. Continuous adjustment of the collected research object’s posture, light, age, and other related characteristic parameters; feedback on the basic model; local differential correction until there was no specific representation or optimization of the general representation, infinitely close to 100% of the animal entity; and the optimization of the basic animal’s model for proportional plotting were performed [14].

To strengthen the classification or ensure that it approximates the actual model, the level of the collected representation dataset was continuously generalized to process specific enhanced representations to eliminate biological representations generated under specific conditions or provide evidence for weakening and correcting specific representations [14].

We used Artrage 6.0 painting software (https://www.artrage.com/, accessed on 4 July 2019), GNU Image Manipulation Program (GIMP) version 2.10 (https://www.gimp.org/, accessed on 8 July 2020), Photoshop 2021 (https://www.adobe.com/, accessed on 13 June 2024), and other software to process pre-selected photo images and adjust parameters, such as color, white balance, and exposure, to the same standard for color-type judgment, and after that, determined the body color of each captured takin by comparing it with a color palette. Moreover, we scaled and drew vector images of different body colors and ages of takins using a 1:10 ratio.

### 2.5. Suitable Habitats

We simulated the distribution range and main habitat size of the two takin subspecies in the study area using a MaxEnt model (3.4.1) with known collected distribution points, including the potential habitat index of 0.5–0.7 and high suitability habitat index of 0.7–1.0. A grid file with a resolution of 1 km^2^ was used for 19 bioclimatic factor variables and 11 other variables (altitude, annual precipitation, solar radiation, annual average temperature, annual maximum temperature, annual minimum temperature, water vapor pressure, wind speed, land cover type influencing factor, human activity influencing factor, and water body). Based on the actual survey of species activity sites, we defined a habitat distribution index of 0.5–1.0 as the actual distribution range of the species. The simulated data were corrected by adding limiting factors to the on-site survey [15,16,17]. In summary, we have compiled 60 locations for Mishmi takins and 54 locations for Bhutan takins to operate the model. 

### 2.6. Population 

We used the Rowcliffe et al. (2008) method to estimate the population size of a species with difficulty in the individual identification of Ips obtained by camera trapping. The method is summarized as follows: species surveyed using camera trapping are assumed to be molecules in a closed space, the area covered by movement within a certain time as molecular linearity and movement distance, and the expected collision rate between molecules within a specific time range as the proportion of the area covered by the movement of all molecules to the total area [17,18,19].

Biologically, the number of collisions y (total number of Ips of individual species) between a relatively stationary molecule (species) and a circular detection area (camera perspective) was determined by the movement speed v of the molecule (species), movement time t (camera workdays), distance r (radius of the camera monitoring area), and molecular density D (species density), as shown in Equation (1): y = 2*rtvD*(1)

Since the monitoring area of the camera is not circular but rather based on a radius, r, and an angle, θ, the fan-shaped area formed by the composition, which is the width covered by the captured molecules (species), is no longer 2r, but the fan-shaped area formed by the angle (π−θ)/2 ≤ γ ≤ π/2, as derived by Rowcliffe et al. in Equation (2):(2)$$y\;=\;\frac{2+\theta }{\pi }rtv\theta$$

According to a study by Rowcliffe et al. on the population density of Chinese water deer, the statistical *t*-value was based on six weeks of daytime camera work (12 h) as the effective number of hours. Equation (3) is derived as follows:



(3)
$$\text{MVC}\;=\;\frac{y}{t}$$



MCV is the annual shooting value, y is the Ips of takins that year, and t is the number of days in a year when cameras were deployed. In Equation (4),
(4)$$D\;=MCV\frac{\pi }{vr\;(2+\theta )}$$

The monitoring angle of the infrared camera, θ, was 0.872 rad (50°), and the camera monitoring radius r was 0.01 km (10 m). On the spot, it was found that the daily activity distance of takins in the sample plot was generally 5–10 km. Daily movement speeds of takins were set at v_1_ (5 km/d) and v_2_ (10 km/d).

### 2.7. Activity Rhythms

The specific time for each independent detection of the two subspecies was calculated, and the kernel density function was used to express the daily activity rhythms of the two takin subspecies. In this analysis, each independent detection was considered a random sampling of the probability of distribution of activity intensity of the two takin subspecies by camera trapping within 24 h. Because of the continuous camera-trapping operation for 24 h, the probability of this species being captured by camera trapping during the active period of the day is consistent. Therefore, the probability density function can be used to characterize the all-day activity rhythms of the two takin subspecies. This analysis was completed using the “overlay” package in R (v3.4) [19,20].

### 2.8. Animal Classification

Hodgson (1850) first obtained a solid specimen from the Mishmi Mountains of southern Tibet, China. Due to the overlapping characteristics of the species, which presented the body shape of *Bos* and the color of *Dorcas*, he incorporated the names of the two genera. He identified a new genus and species: “*Budorcas taxicolor*” (from https://www.itis.gov/, accessed on 10 November 2023. Taxonomic Serial No.: 898763) [7,8].

Subsequently, Edwards (1874) established the subspecies of Sichuan takin (*B. tibetana*) (from https://www.itis.gov/, accessed on 10 November 2023. Taxonomic Serial No.: 898765) based on specimens from Baoxing, Sichuan [5,6]. In the same year, Lydekker identified a smaller and darker takin collected from the border between Xizang and Bhutan as a subspecies of *B. whitei* (from https://www.itis.gov/, accessed on 10 November 2023. Taxonomic Serial No.: 898766), whose origin was Bhutan [6,7,21,22,23,24].

Tomas (1911) designated an independent species (*B. bedfordi*) (from https://www.itis.gov/, accessed on 10 November 2023. Taxonomic Serial No.: 898764) based on specimens obtained by Anderson from the Taibai Mountains in the Qinling Mountains, Shaanxi Province, China. This white or golden takin, known as the Golden takin, originated in the Qinling Mountains [7,8].

Lydekker (1916) believed that in determining the taxonomic status of takins in the regions mentioned above, the Mishmi takin, Sichuan takin, and Qinling takin should be considered three independent species and the Bhutan takin a subspecies of the Mishmi takin. Ellerman et al. (1951) classified *Budorcas* as a genus of species with four subspecies, namely the Mishmi (*B.t. taxicolor*), Bhutan (*B.t. whitei*), Sichuan (*B.t. tibetana*), and Qinling (*B.t. bedfordi*) [23].

The classification of *Budorcas* remains controversial. Because of the considerable differences in body color between Qinling and Sichuan takins, researchers can easily distinguish these subspecies, and ecological research has developed relatively well. However, due to limitations in research conditions and investigation methods, differentiating Mishmi takins from Bhutan takins, distributed in the Eastern Himalayas of China, is difficult, thus preventing further research progress [7,25,26,27].

### 2.9. Ethical Statement

This study was conducted strictly following the guidelines of the relevant ethical committee or organization. The Linzhi City Forestry and Grassland Bureau of the Research Area approved the research protocol, including all pertinent experimental details. We used standard ecological assessment methods in the research process, including a camera-trapping setup, photo sample collection, and data analysis. All applicable institutional and national guidelines for the care and use of animals were followed.

## 3. Results

Of the 1210 independent photographs (Ips) of takins obtained during this camera-trapping survey, 540 were taken during the day, accounting for 44.63% of the total takin Ips, and used for morphological recognition. Two different takin populations, Mishmi takins and Bhutan takins, were recorded (Figure 2 and Figure 3).

### 3.1. Identification Features of Mishmi Takin (B. taxicolor) 

The Mishmi takin has an adult head length of approximately 1.8–2.2 m and a weight of 250–400 kg. The tail is approximately 15 cm long and is usually hidden under thick, long, and fluffy fur. The physical characteristics of this species are similar to those of other takin subspecies. These characteristics include a thick and sturdy body with four limbs. The shoulders are higher than the hips, the tail is short, and the snout is high and curved, resembling that of a sheep (Figure 4). The coat is white-yellow, or golden to reddish-brown, with dark stripes on the back. The shoulder height of a male takin is approximately 120 cm, whereas that of a female takin is approximately 105 cm. The head of a takin is large with an arched muzzle and a wide exposed nose. Both sexes have horns of approximately 64 cm in length. Adult horns have smooth tips and are bent from the base of the head to the sides and twisted back and up with the horns pointing inward. The legs are short, with large, sturdy, and well-developed two-toed hooves. The color of the backs of elderly takins is close to dirty white, with black ridges. The snouts and limbs are black. The juveniles are grayish-brown throughout the body (Figure 2).

### 3.2. Identification Features of Bhutan Takin (B. whitei) 

The head length of an adult Bhutan takin is approximately 1.6–2.0 m, and the weight is 200–300 kg. The overall hair color is light golden-yellow or deep brown, with long-beard hair growing beneath the jaw and neck. Both males and females have thick, smooth horns that bend from the top of the head to the sides and then twist back and up, with the horns pointing inward. The body is thick and sturdy, with four limbs and the shoulders higher than the hips. The tail is short, and the snout is high and curved, resembling a sheep’s. The shoulders are higher than the buttocks, with thick, curved corners on both sides. The color and luster of the fur vary from old to young ages. The body is covered in dirty white fur and nestled in yellow-white fur. Older takins are dark brown, with ridges on their backs and black snouts and limbs. The head and limbs of the juveniles are black, and the rest of the body is covered with grayish-brown fur (Figure 3).

Based on actual investigations, for those lacking experience, the individual morphologies of the two subspecies at a 1:1 comparison involve only subtle differences. Thus, it is difficult to distinguish between the two subspecies using a single or individual comparison feature; rather, it is often easier to distinguish them when there are considerable differences in individual traits or features. When takin populations increase to 100:100 or 10,000:10,000, similar to the differences in humans or school uniforms, small differences in data are amplified and become clear at a glance.

### 3.3. Seasonal Traits 

External and internal variables, such as age, season, temperature, humidity, and geomorphic characteristics, easily influence seasonal traits. The colors of the fur coats of the Bhutan takin and Mishmi takin may differ considerably because of the above factors. The Mishmi takin exhibits dirty white fur in the field, whereas the Bhutan takin exhibits orange-black fur. An adult stable trait refers to a trait that is not easily changed after maturity. The Mishmi takin shows less dirty white fur on the back and abdomen, whereas the Bhutan takin shows a larger orange-black color in the same body area. In summary, using the independent site full-event observation method, it is easy to distinguish between the two takin subspecies via long-term observations based on body color, morphology, and behavior.

According to the survey, the wild survival status of the two takin subspecies is a family comprising one male polygamous-raised offspring. We used an electronic device instantaneous scanning method to obtain a photograph of a family of 47 takins of different sizes, including one adult male, accounting for 2.13% of the population. There were 26 females, which accounted for 55.32% of the population; 9 juveniles 0–1 years, which accounted for 19.15% of the population; and 11 sub-adults 2–5 years, which accounted for 22.40% of the study population (Figure 5). The other type is a sexually mature takin or an old herd of lone bulls abandoned by the group. There are two sources of solitary bulls. First, mature male takins in this or other groups may be expelled from the group by the lead bull and then wander alone until they are strong and capable of occupying a group, or a previous group leader may be defeated by a mature takin and become forced outside the group, which drives the wandering takins to become members of all-male groups.

Second, in animal populations with distinct social group levels, the main male often has a substantially larger body size than the other males in the group based on the premise of food source prioritization and sexual selection [7,8]. Due to evolution and the increasing number of animal species, subtle differences are notable in morphological data, distribution, behavior, and activity patterns.

The Mishmi takin is greater than the Bhutan takin in body shape. In terms of adult body color, the former generally shows dirty white fur, with the base color of the coat a mixture of dirty white and sparse black hair. The latter has a light-orange base and a mixture of black hair. When observed in the wild, Mishmi’s skin appears white or black. The overall color of the Bhutan takin is either orange or black.

### 3.4. Suitable Habitats for the Two Takin Subspecies 

Six of the twelve survey areas recorded animals belonging to the takin species. The lowest and highest altitudes of the recording sites were 582 m and 3500 m, respectively. We captured 4837 Ips of takins, including 1685 and 3152 Mishmi and Bhutan takins, respectively (Table 1). Other recognizable ungulates distributed in the same domain included the Red goral (*Naemorhedus baileyi*), Red Muntjac (*Muntiacus vaginalis*), Gongshan Muntjac (*Muntiacus gongshanensis*), and Fea’s Muntjac (*Muntiacus feae*).

Of the 3365 Ips taken during the day, only one animal was recorded each time. There were 1123 and 2242 Mishmi and Bhutan takins, accounting for 33.37% and 66.63% of the total recognizable photos, respectively.

According to the MaxEnt niche model, the optimal habitat area for Mishmi takins was 2404 km^2^, with a distribution area of 17,314 km^2^. The most suitable habitat area for the Bhutan takin was 4156 km^2^, with a distribution area of 25,006 km^2^.

According to the simulation and survey results, Bhutan takins’ most suitable habitat in the study area is the largest, distributed on both sides of the Yarlung Zangbo River. Moreover, the two takin subspecies are not affected by the river barrier, and there is an area of mutual communication and mixing on both sides of the Yarlung Zangbo River (including the investigation area of Duoxiong Valley; the eastern Nibi Valley was occupied by Bhutan takins, and the western Bixiri Valley by Mishmi takins, which is located on the west bank of the Yarlung Zangbo River). We believe that the variation in the fur color is a product of the mutual penetration of population genes. This genetic exchange between the populations indicates that reproductive isolation did not occur between the two subspecies.

Combining the actual survey and simulated distribution results, we drew the following conclusions: Mishmi and Bhutan takins are distributed on the east and west banks, respectively, with the Yarlung Zangbo River as the boundary. The two takin subspecies occupied a mixed area on the east and west banks of the Yarlung Zangbo River without a strict division. Owing to the lack of the literature on the morphological differences between the two subspecies, most researchers have long been unable to distinguish between Mishmi takins and Bhutan takins, resulting in unclear explanations of the current situation. The MaxEnt niche model showed a habitat index of 0.7–1.0, indicating that the ecological niche differentiation of the two takin subspecies was not obvious and that a high overlap in their core habitats did not occur. The overlap ratio between Mishmi takins and Bhutan takins was 0.4:0.3. When the habitat index was 0.5–1.0, the overlap ratio between Mishmi takins and Bhutan takins was 0.5:0.4. No considerable overlap was observed between the core habitats of the two takin subspecies in the study area. The main distribution area of Mishmi takins is still concentrated in the Mishmi Mountains in southern Medog County. The main distribution area of Bhutan takins is along the Polong Zangbu River (upstream) and in the southern mountainous areas of Milin County.

The silver-white circular arrows in Figure 6 indicate the horizontal migration path of the two subspecies, which was confirmed by actual investigations and camera trapping. The internal driving force behind this horizontal migration may be related to excessive population density within the region and the search for new food sources. The direction of migration may be positively related to the direction of flow of the Yarlung Zangbo River. 

### 3.5. Populations of the Two Takin Subspecies 

Based on the movement speed v_1_ (5 km/d) and v_2_ (10 km/d) parameters of the two subspecies, the densities of the two takin subspecies were calculated as D_1_ and D_2_, respectively (Table 2). No significant difference was observed between Mishmi takins (F = 1.2744, *df* = 1, *p* > 0.05) and Bhutan takins (F = 1.1201, *df* = 1, *p* > 0.05). The statistical analysis revealed no significant difference in population density; therefore, the actual density was based on the average density. The population density of Mishmi takins in the study area was 0.1729 ± 0.0134 takins/km^2^, and the population size was 2995 ± 232 takins. The population density of Bhutan takins in the study area was 0.1359 ± 0.0264 takins/km^2^, with a population size of 3398 ± 660 takins.

### 3.6. Activity Rhythms of the Two Takin Subspecies

During the region’s rainy season from May to October, the takins were active at an altitude of >3500 m.

Using the whole-day nuclear density map, we concluded that both takin subspecies maintained a high activity intensity, presenting a “double peak” pattern in the morning and evening, at 07:00–09:00 and 16:00–20:00, respectively. Owing to minimal human interference in the study area, takin populations maintain a rhythm of activities, such as foraging, playing, and communicating, in the morning and evening. There are obvious concave spots with reduced activity from 10:00 to 12:00 (Figure 7).

The annual activity frequency intuitively presented a “double peak”, with a gradual increase in activity frequency from January to April. During the rainy season (May–October), takin activities gradually increase as the altitude increases (>3500 m); therefore, takin activity was relatively low in areas below 3500 m from May to October. As the temperature rises and the rainy season approaches in low-altitude areas, the living environment of the takins becomes exceptionally difficult, with the animals suffering from mosquito infestations and experiencing relatively high difficulty in obtaining food. During the rainy season, the feeding and living environments in the high-altitude meadow areas were relatively mild, exhibiting seasonal vertical migration habits (Figure 8). The activity range during the rainy season (May–October) involved two natural zones: the alpine cold zone shrub meadow zone (ASL: 3800–4200 m) and the alpine frozen weathering zone (ASL: 4200–4800 m). The dry season (November–April of the following year) covers three natural zones: the mountainous warm temperate mixed coniferous and broad-leaved forest zone (ASL: 2300–2900 m), mountainous subtropical evergreen broad-leaved forest belt (ASL: 1100–2300 m), and low-mountainous tropical rainforest belt (ASL: <1100 m). The transitional zone for seasonal migration is the mountainous cold-temperature dark coniferous forest zone (ASL: 2900–3800 m). This natural zone exhibits considerable extensibility during rainy and dry seasons, providing a solid food source for upward and downward migrations of takins. Relevant research and actual investigations have shown that the spatial migration of takins is mainly influenced by sexual selection and food sources (Figure 9).

## 4. Discussion

With the application of electronic monitoring devices and software technology in wildlife conservation in the past two decades, a large amount of vector data from photos and videos have been generated, which provides the possibility for a correct understanding of wildlife and promotes significant development in this field [1,2,3,4]. Based on previous research, this article uses many photos and videos and utilizes relevant software to finely display the possible differences between the two subspecies of takins. Based on the support of survey data and long-term field investigations, the MaxEnt model provided the distribution range and possible mixed areas of two different takin subspecies in the Eastern Himalayan region and a possible spatial migration model within the region. Comprehensive display of trait data (including individual and population data) and distribution range, activity rhythm, population estimation, and other related basic data for this species filled the research gap of this species.

The study of shape data found significant differences in morphology and fur color between the two subspecies of takin, Mishmi and Bhutan takins; this is difficult and unfriendly for beginners and wilderness identification, and the relevant mechanisms have not been explored and need to be addressed [26,27,28,29,30]. The distribution of the two subspecies in the study area is also vastly different. The Mishmi takin is distributed in the southern part of the region, with a vertical migration range of 500–4500 m. The distribution of the Bhutan takin is generally northward, with a vertical migration range between 1400 and 4500 m. Subspecies 2 has mixed areas in the Yarlung Zangbo River’s lower reaches and a clockwise horizontal migration route. This small spatial difference (horizontal and altitude difference distance) indicates that the species has different coping strategies when dealing with complex terrain and climate environments. Simultaneously, this strategy of coping with the environment has led to mixed phenomena between the two subspecies in some areas. Long-term hybridization indicates no interspecific isolation between the two subspecies, resulting in the mutual infiltration of genes between the two populations and thereby reducing the differences in the species’ morphology. The above possible inference has been supported by reliable evidence obtained through investigation and research, as described in this article.

Due to a lack of detailed observation, it was initially widely believed that animals in nature do not undergo hybridization into species [31,32,33,34,35]. However, as more and more genomic data of species are analyzed, zoobotanists have found that reproductive isolation between species within many taxa is not as strong as imagined, and hybridization events may frequently occur [36,37,38,39]. It is known that multiple animal groups have undergone hybridization events, including insects, fish, birds, canines, non-human primates, and carnivores. These groups have all undergone rapid adaptation radiation, which means a large outbreak of new species in a short period.

The forests around the Yarlung Zangbo River in the study area have been transformed, with rapid urban population expansion leading to the spread of food resources. The migration of wild animals between the two riverbanks may pose survival risks but is worth investigating. One theory suggests that survival choices and food sources exacerbate the formation of hybrid zones on both sides of the river, serving as cradles for newborn animals or graves for aged ones [32,36,37,38,39,40,41]. Gene penetration in hybrids provides new sources of variation, aiding genetic rescue and addressing climate change threats [34]. Another theory suggests that gene exchange from hybridization may diminish differences in evolutionary function and species diversity [42].

The Himalayas, formed by geological phenomena such as continental drift and collision, create conditions for high biodiversity. The interaction of geological structure, erosion, and climatic factors fosters diverse and unique mountain ecological environments. The heterogeneity of these environments drives changes in biological distribution, ultimately leading to population differentiation [43,44,45].

Various technological means are constantly developing, spurring the rapid development of wildlife conservation [1,2]. A correct understanding of complex species is particularly important, whereas an incorrect understanding of species may lead to deviations and delays in conservation policies. The authors believe that a correct understanding of a species and its wide distribution range and a reasonable estimation of its population size are crucial for wildlife conservation and are not easily obtainable.

Indeed, different methods may yield incomplete results in accurately identifying species by rapidly assessing distribution range and population size. The main reasons may be divided into two aspects: The first is the difference in research methods. On the other hand, it may be due to differences in investigation methods; this is the biggest factor currently troubling grassroots forestry workers and decision-makers. However, regardless of the method used, its reasonable reference value is obvious, which is crucial for the accumulation and continued development of basic data on wildlife conservation.

## 5. Conclusions

First, this article utilizes long-term accumulated photo and video vector data from infrared camera traps to finely display the subtle differences between the two subspecies of takin using relevant software. This improved the correct understanding of the species. The distribution range data of two subspecies in the study area and accurate habitat area data were provided using the MaxEnt model combined with actual survey data. The activity rhythms of two subspecies and the migration route models of different subspecies in different seasons were displayed. A reasonable speculation was made that there is minimal difference in trait data between the two subspecies, indicating the existence of gene exchange between them. Second, the above-mentioned reasonable research fills the basic database of the biology and ecology of this species in the region, laying a solid foundation for further research on the species.

Third, current molecular research on large ungulates has confirmed that hybridization leads to gene exchange with populations of species with similar body sizes living in the same domain. Geographical and reproductive isolation appeared to have little effect on species with similar body types. Allelic variations in genetic structure caused by interspecific hybridization infiltration and structural variations in large genome segments trigger phenotypic changes in these species, resulting in variable epigenetic characteristics [5,46,47,48]. In contrast, these structural genetic variations may provide a basis for the species to adapt to the current environment or accelerate the evolutionary history of the species within the region.

These biological and ecological processes constitute a dynamic balance between species formation and extinction. Therefore, mountainous areas may become cradles, museums, or graves. The interaction between long-term climate and terrain evolution results in a higher rate of species formation and unique opportunities for species coexistence and maintenance, ultimately resulting in extremely high biodiversity in mountainous areas. Therefore, biodiversity in mountainous areas results from long-term species evolution and the combined effects of multiple ecological processes [44,49]. Since the Quaternary, climate change has driven cyclical changes in habitat connectivity, triggering species formation in certain groups. For example, during the warm and humid interglacial period, vegetation zones moved upward (to high-altitude areas), leading to population geographic isolation and gene divergence. During the ice age, when temperatures decreased, vegetation zones moved downwards (to low-altitude areas), causing contact between previously geographically isolated populations. During this period, founder effects, disruptive selection, and character displacement may have occurred, ultimately forming exotic species. The theoretical basis for this formation conforms to the “competitive exclusion theory”, meaning that complete competitors cannot coexist [44,45,46].

As it is necessary to assess the impacts of species restoration and conservation, accurate species knowledge is crucial for assessing population size and distribution range [47]. Because of the extremely abundant floral and faunal resources in the Eastern Himalayan region of China, which is limited to areas such as tropical seasonal rainforests and mountainous areas, it is difficult to investigate and clarify the species and animal populations in the region. With the continuous development of scientific and technological methods and the accumulation of data by scientific and technological workers over the years, this problem will soon be solved [1,2].

Studies have shown that multiple pressure factors, including climatic and human factors, can cause changes in species populations and habitats. These typically present complex and effective combinations that enhance or reduce their respective impacts on such populations and habitats to varying degrees [48]. In the short term, the pressure exerted by human factors on individual species determines the survival of the species. Short-term, small-scale human activities determine the survival of species populations. In contrast, long-term, large-scale climate change may be the first issue to be addressed for the species’ overall survival.

Ecological species assessments that include a reasonable population size and distribution range are beneficial. They can help policymakers address the impact of climate change on species and develop relevant conservation strategies to ensure the well-being of all humanity [49]. Moreover, it is beneficial for humans to fully understand the survival status of species in an objective world.

## Figures and Tables

**Figure 1 animals-14-02426-f001:**
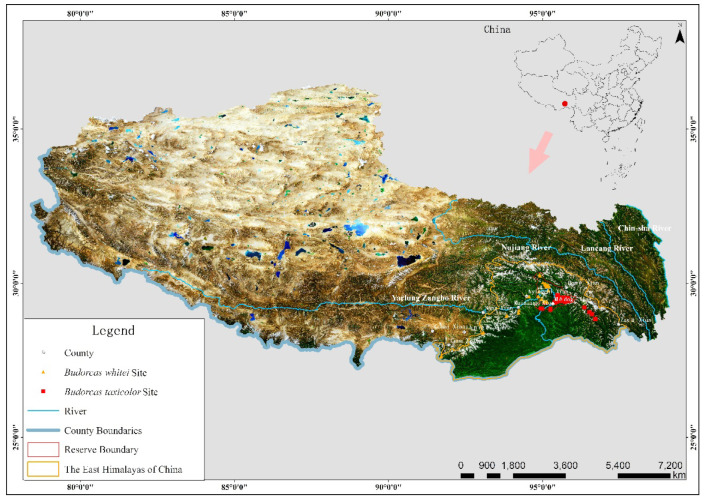
Study area and camera-trap sites. The figure was made with ArcGIS tools (ArcMap 10.7, https://www.esri.com/en-us/arcgis, accessed on 21 March 2019), and the data source used in this figure was downloaded from https://data.tpdc.ac.cn, accessed on 19 May 2022.

**Figure 2 animals-14-02426-f002:**
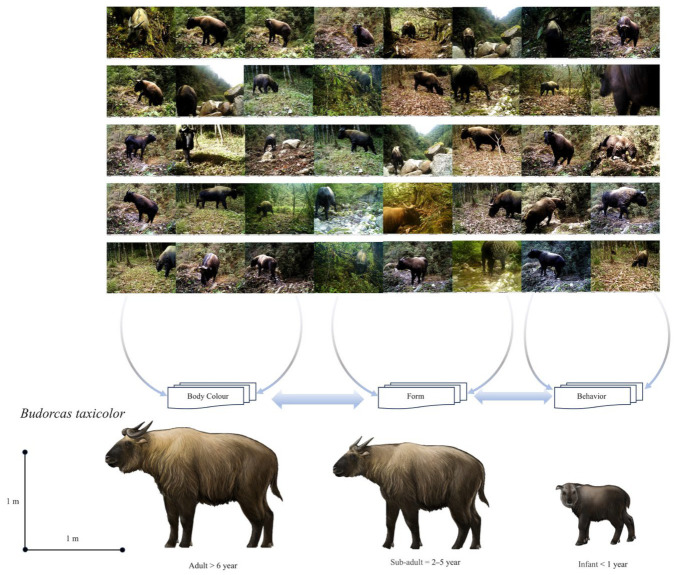
Mishmi takin (*B. taxicolor*) from Eastern Himalayas of China.

**Figure 3 animals-14-02426-f003:**
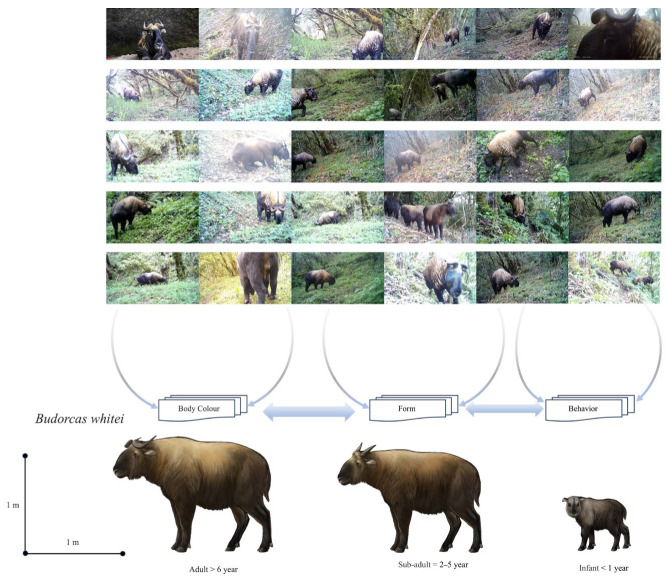
Bhutan takin (*B. whitei*) from Eastern Himalayas of China.

**Figure 4 animals-14-02426-f004:**
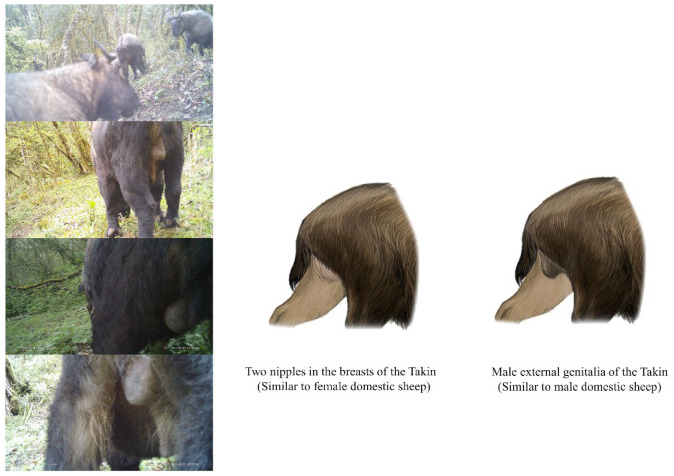
Genital patterns of the two subspecies of takins.

**Figure 5 animals-14-02426-f005:**
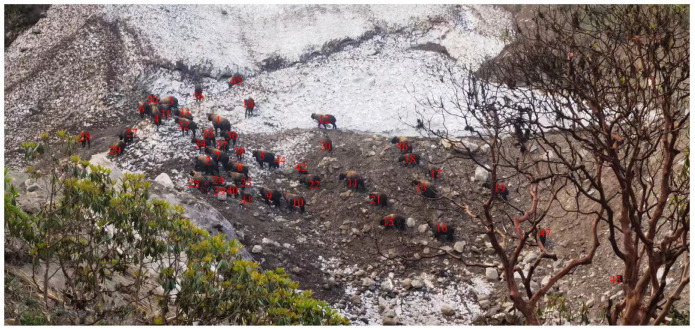
The instantaneous scanning method was used to obtain cluster photos for interpreting actual sample line surveys (47 Mishmi takins are present; this photo was taken in September during the rainy season at an elevation of 4350 m).

**Figure 6 animals-14-02426-f006:**
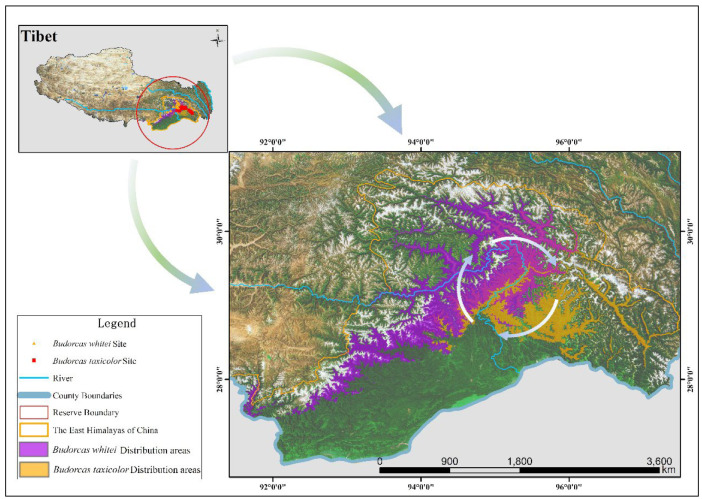
Distribution areas of the two takins (the silver-white circular arrow indicates the possible horizontal migration of the two subspecies in the area during actual investigation). The figure was made with ArcGIS tools (ArcMap 10.7; https://www.esri.com/en-us/arcgis, accessed on 21 March 2019), and the data used in this figure were downloaded from https://data.tpdc.ac.cn, accessed on 19 May 2022.

**Figure 7 animals-14-02426-f007:**
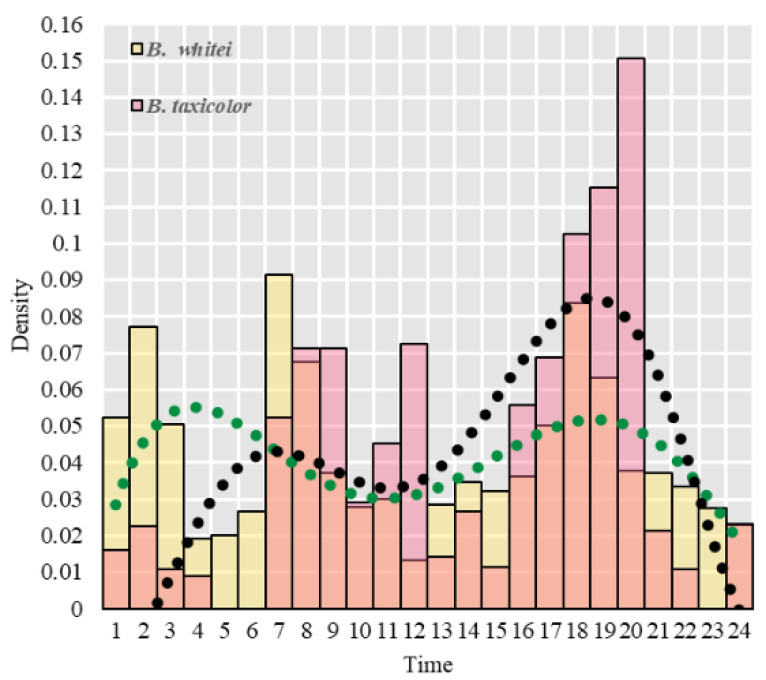
Activity intensity of two subspecies of takin at different periods every day.

**Figure 8 animals-14-02426-f008:**
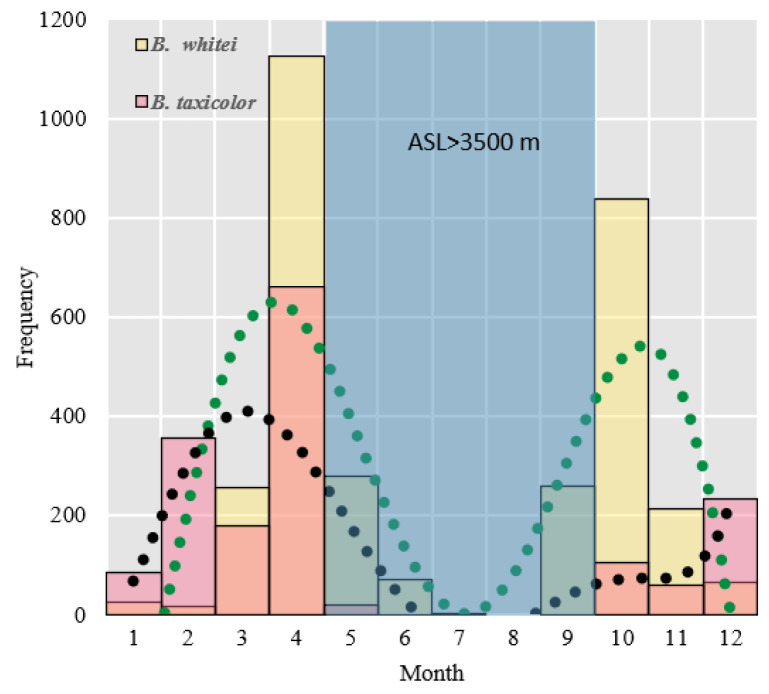
Activity intensity of the two takin subspecies in different months (ASL: altitude above sea level).

**Figure 9 animals-14-02426-f009:**
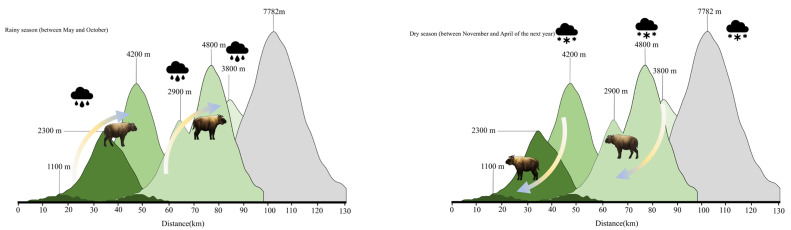
Model diagrams of the seasonal migration of takins. During the rainy season, from May to October, they migrate to an altitude of ≥3500 m. During the dry season from November to April of the following year, their activity occurred at an altitude of <3500 m. Mishmi takins and Bhutan takins were active at altitudes of 500–4500 m and 1400–4500 m, respectively.

**Table 1 animals-14-02426-t001:** Camera-trapping efforts and the number of independent photographs of *Budorcas* in the 12 survey areas from 2013 to 2022.

Survey Areas	Number of Camera Stations	Elevation Range	Number of Camera-Days	Number of Photographs	Number of Independent Photographs of *Budorcas*	
					*B. taxicolor*	*B. whitei*
Bixiri Area	11	2235–3479 m	2794	12,294	832	
Nibi Area	13	2256–3509 m	2354	13,892		648
South Bank of the Yarlung Zangbo River Area	10	582–668 m	1880	16,751		
Uma Mountain Area	6	1751–3145 m	1374	6942	3	
Raj Mountain Area	8	1631–2086 m	1968	8467		
DanGeZhuo Area	3	954–1434 m	630	684		
GeDang Ditch Area	36	2230–4470 m	8691	10,713	843	
MeiYuLunBa Area	2	1751–2315 m	294	5172		
XiGong River Area	6	1124–1590 m	1080	6532		
DeYang Ditch Area	8	815–1294 m	1360	6359	7	
North of the Grand Canyon	31	3650–4700 m	5580	4578		2504
DeErGong Area	80	1750–2890 m	30,500	11,980		
Total	214	582–4700 m	58,505	104,364	1685	3152

**Table 2 animals-14-02426-t002:** *Budorcas* population density was estimated using camera trapping.

*Budorcas*	Ips	Camera-Days	MCV	D_1_	D_2_	/x ± SE
*B. taxicolor*	1685	1238	1.36	0.1738	0.1720	0.1729 ± 0.0134
*B. whitei*	3152	1594	1.98	0.1429	0.1289	0.1359 ± 0.0264

Note: D_1_ = density 1 (individuals/hm^2^); D_2_ = density 2 (individuals/hm^2^); Ips, independent photos; MCV is the annual shooting value, as in Equation (3); SE is the standard deviation of the sampling distribution of the sample mean, indicating the stability of the sample mean.

## Data Availability

All data generated or analyzed during this study are included in this published article (and its Appendix A).

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
