# Peer review of "Identification of Individuals of Two Takin Subspecies Using Biological and Ecological Criteria in Eastern Himalayas of China"

_animals, 2024, doi:10.3390/ani14162426_

Round 1
Reviewer 1 Report
Comments and Suggestions for Authors
Dear Authors,
The manuscript certainly touches on an interesting topic. Issues of identifying animal subspecies will always be relevant. But any researchers face many difficulties in distinguishing one animal taxon from another. Therefore, the manuscript contains extremely important scientific data, but it cannot be published in this form of the manuscript. The text of the manuscript in different chapters is mixed and should be moved to the appropriate chapters. The title of the manuscript does not quite correctly reflect its content. In many methodological aspects, the authors leave out important information. It should be added. The discussion needs to be completely rewritten as it is too theoretical and not directly based on the discussion of the takin research results. Some paragraphs may be moved to the Introduction. Some parts need to be shortened because they do not carry a semantic load within the framework of this study. In the manuscript, the authors provided extremely insufficient information about the molecular genetic study of takins. It is worth dwelling on this in more detail when comparing our own results of identifying takins using different criteria. After all comments have been eliminated, the manuscript can be reconsidered.

Author Response
Response to Reviewer 1 Comments
Point 1: The title of the manuscript does not exactly correspond to its content, since it is primarily about identifying individuals of two subspecies on the basis of biological and ecological criteria. Therefore, I recommend the following title for the manuscript:
«Identification of Individuals of Two Takin Subspecies Using Biological and Ecological Criteria in the Eastern Himalayas of China».
Response 1: Line 2. We are more than happy to accept your suggestions. We changed the title to " Identification of Individuals of Two Takin Subspecies Using Biological and Ecological Criteria in the Eastern Himalayas of China."
Point 2: Scientists are increasingly using various new methods of identification (acoustic, photographic observation) and monitoring of animals, depending on the characteristics of their lifestyle (Andreychev, 2018; Xie and Yu, 2023).
Response 2: Line 58. We have made corrections based on the reviewer's comments and added references.
Point 3: Animal classification should be moved to Materials and Methods to clarify the history of taxonomic status studies.
Response 3: Line 85. We will adjust the classification of the two subspecies to section 2.8 of the Research Methods and Materials section.
Point 4: If you are writing here about molecular studies of takins, then the Discussion should compare your results with these methods.
Response 4: Line 113. What we want to express is that although molecular evidence has been used to combine the Mishmi takin and Bhutan takin in the study area, it is still necessary to have a correct understanding of the two types of takins and their ecological habits from an ecological and macroscopic biological perspective.
Point 5: The aim of the study and the hypothesis are missing here. Therefore, the manuscript should be revised in this part.
Response 5: Line 125. There are still many questions, including the differences in appearance, population size, and distribution range between the two takins in the region.
Point 6: Materials and Methods need to be moved here. The results should follow the research methods statement.
Response 6: Line 128. We have made corrections based on the Reviewer's comments.
Point 7: It is better to transform this descriptive part of the text into the form of identifying keys of two subtypes, based on the principle of comparing similar evaluation criteria.
Response 7: Line 140. We adjusted the diagram and description of the species, presenting and describing them individually one by one. Finally, we summarized the commonalities of sexual organs.
Point 8: How many families were registered in total? How many individuals are there on average in each family? It is necessary to provide limits for varying the number of individuals in families from minimum to maximum.
Response 8: Line 187. Compared to panoramic family group photos, they are rare, and their survey methods are limited. This image is an extremely rare panoramic shot, attempting to illustrate the composition of different age groups within a family group. This shows the way and strategy of species survival in the wild. And the investigation time shows the seasonal migration of this species.
Point 9: This is already a Discussion.
Response 9: Line 192. We believe it is more reasonable here based on the inferential results described above.
Point 10: What is the frequency of camera recordings of these species compared to the takins you study?
Response 10: Line 215. Other articles have been published, and we intend to express the richness of ungulate species in the study area.
Point 11: This is a Discussion. It must be proven by relevant research. This is not addressed in the Materials and Methods.
Response 11: Line 283. Our long-term field investigation experience has effectively inferred the activity rhythm chart.
Point 12: The discussion needs to be completely rewritten as it is too theoretical and not directly based on the discussion of the takin research results. Some paragraphs may be moved to the Introduction. Some parts need to be shortened because they do not carry a semantic load within the framework of this study. In the manuscript, the authors provided extremely insufficient information about the molecular genetic study of takins. It is worth dwelling on this in more detail when comparing our own results of identifying takins using different criteria.
Response 12: Line 319. We have written a discussion section from the new book, focusing on the intersection of distribution and the possible gene exchange that may lead to minor changes in trait data differences.
Point 13: Move higher in the text of the manuscript to Results.
Response 13: Line 444. We moved the section on methods and materials.
Point 14: How far were the areas from each other? What principle was decisive when choosing areas?
Response 14: Line 498. We have continuously monitored the research area for 10 years, with over 200 infrared cameras installed. Due to the mountainous nature of the research area, some areas cannot be reached by manpower. We strive to make up for this by conducting long-term surveys. The reason for conducting surveys in inaccessible areas is that animals are active, and the distance between each camera is greater than 2 kilometers. The principle of all these settings is to minimize the error value of blank areas as much as possible.
Point 15: How were the cameras placed at the areas? How many cameras were located at each of the twelve areas? At what height were the cameras placed? Was camera camouflage used?
Response 15: Line 502. Human placement, selecting different areas, landforms, and landscapes. Refer to Table 1 for details. The deployment height is generally between 0.5–1.0 m and may vary slightly depending on the deployment location. Not disguised.
Point 16: This subchapter should not stand alone. It is advisable to combine it with the previous subchapter.
Response 16: Line 547. We believe separating the camera deployment method from the species identification method is more appropriate.
Point 17: You already wrote about this above. Should not be repeated. It is better to combine text that communicates similar methodological approaches.
Response 17: Line 573. We accepted the suggestion and deleted the paragraph.
Point 18: This part should be written in detail. I do not miss many methodological details on the biological rhythms of animals. In particular, it is necessary to indicate the mechanism for identifying the activity of the studied animals by cameras after identifying predators or competitors on camera. This is important from the standpoint of eliminating the subjectivity of research.
Response 18: Line 635. The current methods for capturing animal statistical activity rhythms using infrared cameras and simulating suitable habitats using MaxEnt models are relatively mature. Due to space limitations, the above two methods cannot be elaborated in detail.

Reviewer 2 Report
Comments and Suggestions for Authors
The authors explore the diversity of takins in this unique region of China, highlighting the need for further research to understand their ecological significance. The study employs camera trapping technology across diverse altitudes and vegetation types, allowing for a thorough assessment of takin species in their natural habitat. The research contributes valuable data to understanding the species in the Eastern Himalayas, a recognized biodiversity hotspot, which is crucial for conservation efforts.
While the study provides valuable insights and is an important contribution considering the little published on takins. However, the manuscript is compiled incorrectly, and the English needs to be corrected for scientific soundness. The Methods section is at the end; until then, reading was guesswork! Please move the section to its correct place after the Introduction and before the Results. Also, the AI used for identification must be explained in more detail.
The paper briefly mentions human influence on species survival but could benefit from a more detailed analysis of specific human activities (e.g., habitat fragmentation, poaching) and their direct impacts on takin populations and movements/migrations. Expanding the discussion to include interactions with other species and the overall ecosystem dynamics could provide a more holistic view of the takin's role in their habitat and the implications for biodiversity conservation. While the study contributes to conservation knowledge, it could be strengthened by providing specific, actionable recommendations for policymakers and conservationists based on the findings, ensuring that the research translates into practical conservation strategies.
Illustrations are very instructive and a good touch to the manuscript.
Combine the first two paragraphs of the introduction.
Results—How did you calculate head and tail length and weight from your photos? Or is it from the literature? If so, this is not the place for it. Here, you should present your results only.
That also leads to the question of how you sex them from the pictures other than their genitals. You say horns are the same size and that shoulder height is greater in males, but how do you gauge that from the pictures?
Pg 7, line 216 – give Latin names of all species when first mentioned.
Fig 5, 6 caption – please mention that these are averages and the sample sizes.
Paragraph beginning line 289: Basic data like sample sizes are not mentioned. You have to be transparent in your data and analyses.
Discussion: This section needs to be rewritten. It is wholly irrelevant and does not discuss its own data but goes on and on about taxonomy and other subjects. You have to be focused on your subject and study species.
Comments on the Quality of English Language
The English needs to be corrected for the scientific soundness and flow of the paper.
Author Response
Response to Reviewer 2 Comments
Point 1: Combine the first two paragraphs of the introduction.
Response 1: We are very pleased and have accepted your suggestion.
Point 2: Results—How did you calculate head and tail length and weight from your photos? Or is it from the literature? If so, this is not the place for it. Here, you should present your results only.
Response 2: Based on many references and practical investigations, we have provided an interval value to improve our comprehensive understanding of this species. Not just the form data.
Point 3: That also leads to the question of how you sex them from the pictures other than their genitals. You say horns are the same size and that shoulder height is greater in males, but how do you gauge that from the pictures?
Response 3: Based on a massive database of photos and videos and comparing relevant references and historical data, we have obtained the morphological characteristics of this species that are close to reality.
Point 4: Pg 7, line 216 – give Latin names of all species when first mentioned.
Response 4: We have accepted your suggestion.
Point 5: Fig 5, 6 caption – please mention that these are averages and the sample sizes.
Response 5: The sample size of independent photos is shown in Table 2.
Point 6: Paragraph beginning line 289: Basic data like sample sizes are not mentioned. You have to be transparent in your data and analyses.
Response 6: The sample size of independent photos is shown in Table 2.
Point 7: Discussion: This section needs to be rewritten. It is wholly irrelevant and does not discuss its own data but goes on and on about taxonomy and other subjects. You have to be focused on your subject and study species.
Response 7: We have rewritten a discussion section in the book, focusing on the intersection points of distributions and gene exchanges that may lead to minor differences in trait data. The possibility of species hybridization and the gradual disappearance of trait data caused by hybridization.

Round 2
Reviewer 1 Report
Comments and Suggestions for Authors
Dear Authors,
I am pleased with the correction of the manuscript. Corrections and additional data have been added to the manuscript. A reasonable speculation was made that there is minimal difference in trait data between the two subspecies, indicating the existence of gene exchange between them. The distribution range data of two subspecies in the study area and accurate habitat area data were provided using the MaxEnt model combined with actual survey data. The review of the study results was selected accordingly, as were the statistical methods used to analyze it. The article takes into account comments on the methodology. The analysis and conclusion for each chapter are sufficient and unobjectionable. References to literature sources have been corrected. The results of previous studies by other authors are taken into account. I recommend it for journal Animals.
Author Response
Point 1: Dear Authors,I am pleased with the correction of the manuscript. Corrections and additional data have been added to the manuscript. A reasonable speculation was made that there is minimal difference in trait data between the two subspecies, indicating the existence of gene exchange between them. The distribution range data of two subspecies in the study area and accurate habitat area data were provided using the MaxEnt model combined with actual survey data. The review of the study results was selected accordingly, as were the statistical methods used to analyze it. The article takes into account comments on the methodology. The analysis and conclusion for each chapter are sufficient and unobjectionable. References to literature sources have been corrected. The results of previous studies by other authors are taken into account. I recommend it for journal Animals.
Response 1: Dear Reviewer, Thank you very much for your detailed review of our manuscript and for your positive evaluation. Your feedback is extremely valuable to us, and your suggestions and comments have not only encouraged us but also helped us further improve the quality of our research. We have carefully considered all your suggestions and have made the corresponding revisions to the manuscript. We greatly appreciate the time you took to review our work. Once again, thank you for your valuable time and professional insights. We look forward to receiving further feedback on our final version.

Reviewer 2 Report
Comments and Suggestions for Authors
This is a review of a previously submitted manuscript.
Suggest shortening the title to: Identification of Individuals of Two Takin Subspecies Using Biological and Ecological Criteria in the Eastern Himalayas of China
Lines 80 – 81: “…..the two subspecies are cross-distributed sympatric in the …”
Line 173: “ … number of times sessions.”
Lines 181-182: Non-identical cameras can be used to validate animal body color comparisons. – Unclear what this sentence means. Also, this kind of a determination requires a reference to substantiate it.
Line 209: “ …. corpses, and solid bodies, were used …” - what are solid bodies?
Line 212: “Here, we We applied self-trained machine …….. “
Line 221: “the animals entity, ……."
Line 328: “ …… (Figure 2, and Figure 3).”
Line 249 you say “Owing to the relatively single color change in takins,…” but then in Lines 312, 313 you write “Because of the considerable differences in body color between ………..” and then in line 426 you say “We believe that the variation in the fur color… “– Which of them is it? Are they similar or not?
Line 390: “ ……. which drives the wandering takins to become members of a lone takin.” Rewrite so it makes sense.
Lines 395 – 397: The claim you make in this paragraph must be substantiated by appropriate references.
Line 519: “….areas of two different antelope subspecies …” – antelope? Where did this come from?
Comments on the Quality of English LanguageThe language is still not up to par, and I have made a few suggestions. A lot more editing is needed to correct the problems.
Author Response
Point 1: This is a review of a previously submitted manuscript. Suggest shortening the title to: Identification of Individuals of Two Takin Subspecies Using Biological and Ecological Criteria in the Eastern Himalayas of China.
Response 1: Line 1. We are more than happy to accept your suggestions. We changed the title to " Identification of Individuals of Two Takin Subspecies Using Biological and Ecological Criteria in the Eastern Himalayas of China.".
Point 2: Lines 80 – 81: “…..the two subspecies are cross-distributed sympatric in the …”.
Response 2: The word ' sympatric ' accurately describes the cross distribution of species, and we have accepted this suggestion.
Point 3: Line 173: “… number of times sessions.”
Response 3: We accept the suggestion to modify the sentence to “... number of times sessions.”
Point 4: Lines 181-182: Non-identical cameras can be used to validate animal body color comparisons. – Unclear what this sentence means. Also, this kind of a determination requires a reference to substantiate it.
Response 4: We accept the suggestion to modify the sentence to “Non-identical cameras can be used to verify the body color, age, size, and other characteristics of the animals being compared.”.
Point 5: Line 209: “ …. corpses, and solid bodies, were used …” - what are solid bodies?
Response 5: We accept the suggestion to modify the sentence to “Other materials, such as fur, corpses, and animal feces, were used as supporting evidence.”.
Point 6: Line 212: “Here, we We applied self-trained machine …“.
Response 6: “self-trained machine learning” involves the continuous identification and analysis of various photos, videos, objects, animal feces, and other data. The goal is to minimize subjective biases and narrow the gap between human cognition and objective reality. The following paragraph provides a detailed explanation of this method.
Point 7: Line 221: “the animals entity, …….".
Response 7: The animal entities referred to here are proportionally accurate models, representing the animals as they exist in objective reality.
Point 8: Line 328: “ …… (Figure 2, and Figure 3).”
Response 8: It has been verified to be accurate.
Point 9: Line 249 you say “Owing to the relatively single color change in takins,…” but then in Lines 312, 313 you write “Because of the considerable differences in body color between ………..” and then in line 426 you say “We believe that the variation in the fur color… “– Which of them is it? Are they similar or not?
Response 9: Lines 312 and 313 indicate that the body color changes in the Qinling and Sichuan takins are significant, whereas those in the Bhutan and Mishmi takins are minimal. The latter two subspecies exhibit only minor and uniform changes in body color.
line 426 We speculate that this minimal variation may be due to gene exchange between the two subspecies populations or insufficient reproductive isolation, which allows them to interbreed in specific environments. The frequent exchange of these genes results in the two subspecies having very similar fur color and appearance, making them difficult to distinguish.
Point 10: Line 390: “……. which drives the wandering takins to become members of a lone takin.” Rewrite so it makes sense.
Response 10: We have revised the specific term to "all-male group of takins" and incorporated it here.
Point 11: Lines 395 – 397: The claim you make in this paragraph must be substantiated by appropriate references.
Response 11: After thorough investigation and extensive review of references for accurate citation and validation, we have included the works of Professor Wu Jiayan, as cited in references 7 and 8.
Point 12: Line 519: “…areas of two different antelope subspecies …” – antelope? Where did this come from?
Response 12: We have corrected the term "antelope" to "takin" throughout the entire text, acknowledging this oversight. The animal noun has been updated accordingly.
